# Structure-Based Designing, Solvent Less Synthesis of 1,2,3,4-Tetrahydropyrimidine-5-carboxylate Derivatives: A Combined In Vitro and In Silico Screening Approach

**DOI:** 10.3390/molecules26154424

**Published:** 2021-07-22

**Authors:** Uzma Arshad, Sibtain Ahmed, Nusrat Shafiq, Zaheer Ahmad, Aqsa Hassan, Naseem Akhtar, Shagufta Parveen, Tahir Mehmood

**Affiliations:** 1Department of Chemistry, Government College Women University Faisalabad, Faisalabad 38000, Pakistan; uzmaarshad167@gmail.com (U.A.); aqsahassan92@gmail.com (A.H.); shagufta_organic@yahoo.com (S.P.); 2Scripps Institution of Oceanography, University of California San Diego, La Jolla, CA 92093, USA; 3Department of Chemistry, University of Wah, Rawalpindi 47000, Pakistan; dr.zaheer.ahmad@uow.edu.pk; 4Department of Chemistry, Government Sadiq College Women University, Bahawalpur 63000, Pakistan; drnaseem@gscwu.edu.pk; 5Institute of Biochemistry and Biotechnology, University of Veterinary and Animal Sciences, Lahore 54000, Pakistan; tahir.mehmoodbiochem@uvas.edu.pk

**Keywords:** pyrimidine, solvent less, in silico, in vitro, QSAR, DFT, MTT, molecular docking

## Abstract

Objective: In this study, small molecules possessing tetrahydropyrimidine derivatives have been synthesized having halogenated benzyl derivatives and carboxylate linkage. As previously reported, FDA approved halogenated pyrimidine derivatives prompted us to synthesize novel compounds in order to evaluate their biological potential. Methodology: Eight pyrimidine derivatives have been synthesized from ethyl acetoacetate, secondary amine, aromatic benzaldehyde by adding catalytic amount of CuCl_2_·2H_2_O via solvent less Grindstone multicomponent reagent method. Molecular structure reactivity and virtual screening were performed to check their biological efficacy as an anti-oxidant, anti-cancer and anti-diabetic agent. These studies were supported by in vitro analysis and QSAR studies. Results: After combined experimental and virtual screening **5c**, **5g** and **5e** could serve as lead compounds, having low IC_50_ and high binding affinity.

## 1. Introduction

Heterocyclic compound chemistry has revolutionized the modern and versatile medicinal chemistry [1]. These heterocyclic compounds have a very expansive effect on human life and influence heterogeneous fields like medicine, polymer science, agronomy and various chemical industries [2]. These heterogeneous compounds, constituted of dihydropyrimidinones (DHPMs), have gained the much attention after the discovery of DNA and RNA bases having purines and pyrimidines as core group [3]. DHPMs mode of action is evident from the formation of H-bonding along with nucleotides like uracil, thymine, cytosine, adenine and guanine in both DNA and RNA [4]. This class of heterocyclic synthetic compounds contains many naturally occurring therapeutic drugs like quinine, emetine, dibucaine, morphine and reserpine, as shown in Figure 1 [5,6]. 

DHPMs become part of many medicinal and therapeutic agents which interact with DNA and RNA of the infected cells and terminate the cell division of infected cells by seizing DNA and RNA biosynthesis of these cells, as these DHPMs are structurally related to nucleotide bases of infected cells [7]. Pyrimidine scaffold is of great interest for researchers and versatile capability of potentials. Several biologically active functional moieties contributed to the potential of pyrimidine derivatives. Based on the biologically active functional groups, currently synthesized Pyrimidine-scaffold are shown in Figure 2 [8,9]. Due to this captivating biological and pharmaceutical utilizations, chemists are generally interested in the structural modification of the basic nucleus of DHPMs. Regarding the above discussion we have synthesized a series of DHPMs having pyrido[2,3-*d*] pyrimidine structural framework.

Generally, in the past conventional methods for the synthesis of heterocyclic compounds were applied which was less efficient, time consuming and provided less yield [10]. Following the inception of the green chemistry approach in 1991, it is highly recommendable to synthesize the compounds according to the principles of green chemistry [11]. Therefore, in order to minimize these losses, a fuel efficient, innocuous, having no byproduct or side reactions, eco-friendly Grindstone technique has been employed [10]. In medicinal chemistry, fluorination of heterocyclic compounds has much impact on drug stability as in the case of drug absorption, distribution and mechanism. Moreover, owing to the small size, greater electrostatic interactions due to increased binding affinity, less polarizability, and high dipoles C-bond act as good hydrogen bonding recipient and are better involved in resonance [12]. Biological aspects of pyrimidine derivatives containing fluorine as substituent have been advent from its commercially available drugs as these drugs are included in the FDA approved drug list of 2018, shown in Figure 3 [13]. In accordance with the above discussion, mono substituted fluoropyrimidine derivatives have been synthesized.

## 2. Results and Discussion

We have reported the synthesis of tetrahydrpyrimidine derivatives from the reaction mixture of urea/thiourea, substituting benzaldehydes and ethyl acetoacetate utilizing CuCl_2_·2H_2_O as catalyst. Under these conditions, excellent yield and rapid products were obtained. Moreover, this synthetic method was advantageous over the conventional methods owing to its less solvent, economic, less time consuming and possessing single step properties [14,15].

Targeted synthesized compounds were characterized to confirm their structures through the NMR spectral data (Spectra in Appendix A).

The ^1^H- and ^13^C-NMR spectral data of all compounds **5a**–**h** displayed characteristic peaks by which we were able to identify the corresponding compounds (Table 1 and Table 2).

### 2.1. Computational and Experimental Evaluation

#### 2.1.1. Rationalization of Biological Activities by IC_50_ Values

A series of synthetic compounds **5a**–**5h** were subjected to radical scavenging antioxidant, alpha-amylase inhibitory anti-diabetic and MTT cytotoxicity assay. It is manifested from the activity results that all our target molecules **5a**–**h** were observed to possess good to moderate activity in comparison to standards used. These compounds have IC_50_ values in the range of 6.261–2358 M for scavenging radicals, 6.539–11.27 µM for the enzymatic inhibition of alpha-amylase and 5.351–18.69 µg/mL for suppressing the cytotoxicity of HepG2 cell line. Figure 4, Figure 5 and Figure 6 represents the graphical data of IC_50_ values obtained by GraphPad Prism 8. As it was evident from rational drug design scheme shown in Figure 7, Figure 8 and Figure 9 that the synthetic compound is comprised of pyrimidine derivative along with oxo and thio groups, ester linkage with methyl moiety, benzyl ring with halogen group consisting of fluoro- and chloro-substitutions at ortho, meta and para position with respect to pyrimidine ring moiety. All these moieties play important role in regulating the biological potential, especially the position of halogenated derivatives greatly affect the biological potential of these compounds **5a**–**5h**.

According to limited structure activity relationship, all the *meta* substituted chloro and fluoro derivatives on benzyl ring attached to pyrimidine moiety showed good inhibitory concentration values in comparison to other ortho and para substitution. Compound **5c** (IC_50_ = 6.261 µM) having fluoro substitution at *para* position with respect to pyrimidine moiety having oxo group showed best radical scavenging activity [16]. Compound **5g** with (IC_50_ = 6.539 µM) having *meta* substituted chloro benzyl derivative attached with pyrimidine ring with oxo linkage showed highest inhibitory potential for alpha amylase of all synthesized compounds which is attributed to its electron withdrawing effect [16]. Overall conclusion showed by Figure 8. Cytotoxicity evaluations lead us to the conclusion that compound **5e** with *meta* substituted fluoro derivatives having thio linkage possessed the highest IC_50_ compared to control, as shown in Figure 10. All this structure activity relationship provided us with the conclusion that meta substitution of halogenated derivatives with either oxo or thio linkages provided promising results after in vitro analysis [16,17,18,19,20].

#### 2.1.2. Molecular Docking Analysis for Antioxidant Activity

Molecular docking studies revealed the best fit modes of compounds **5a**–**h** by docking with tyrosine kinase. Results obtained after docking analysis provide us with promising compound **5c**, based on the best binding affinity value in comparison to standard (all data in Appendix A) [21]. 3D and 2D interface of best fit poses as ligand into receptor site of tyrosine kinase were visualized in Figure 10A–E [22,23,24,25].

#### 2.1.3. Molecular Docking Analysis for Anti-Diabetic Activity

The most favorable and best fit binding modes of **5a**–**h** with glucokinase showed that their docking score range from −129.805 to −98.995 kcal/mol, while standard drug docking score is −95.8957 kcal/mol. Glucokinase binding pocket showed best affinity with chloro-substituted compound **5g,** which revealed that compound **5g** has the capacity for enzymatic activity inhibition [24,26] (Figure 11A–E).

Compound **5g**, owing to its oxo substitution, has two hydrogen bonding interactions, namely from the side 3-NH and with oxygen of ester linkage with residues such as Asp409 and Ser411. MolDock score and number of hydrogen bonding interactions of **5g** is in accordance with lowest IC_50_ value of this compound (all data in Appendix A) [24].

#### 2.1.4. Molecular Docking Analysis for Anti-Cancer Activity

Compounds **5a**–**h** showed significantly better MolDock scores than standard 5-fluorouracil when docked with human serum albumin (3B9L), indicating a good binding affinity (all data in Appendix A). Results of docking revealed that compound **5c** shows the best MolDock score of −126.322 kcal/mol. Compound **5c** forms hydrogen bonds from oxygen of ester linkage with Ser454 and Arg197, respectively. Oxygen of ester linked carbonyl form hydrogen bonds with Lys199 and Arg197 shown in Figure 12A–E [24,27].

### 2.2. Redocking and RMSD Calculation

The average RMSD descriptor of all the docked compounds were calculated to corroborate their stability in biological system [28,29,30]. As the justified threshold for RMSD calculation is 2 Å, its mean average RMSD value of all the complexes less than this threshold provides us with the proof that our docking process is validated, as having RMSD for anti-cancer docking protocol 2.0 Å, for anti-diabetic 1.35 Å and for anti-oxidant 0.71 Å [31,32,33,34]. Redocked ligand and co-crystal structure’s native ligand superimposed images are shown in Figure 13A–C [35,36,37].

### 2.3. DFT Studies in Scope of Quantitative Structure Activity Relationship (QSAR) and Computational Description

#### 2.3.1. Frontier Molecular Orbital Analysis (FMO) along with Optimized Structures

Contour diagrams of HOMO LUMO are significant for the determination of chemical reaction mechanism. For this purpose, these parameters must be calculated precisely [38]. Contour diagrams of FMOs comprising the HOMO (highest occupied molecular orbitals) and the LUMO (lowest unoccupied molecular orbitals) in studied compounds are calculated and optimized geometry of structures along with numbering system, and the vector of the dipole moment is shown in Figure 14A,B and Table 3.

According to Table 3, the first computational parameter is E_Homo_. High value of Homo predicts that the compound has good electron donor ability, and can give electrons to the appropriate acceptor molecule, resulting in good biological activity associated with increasing value of E_Homo_ [39]. Thus, activity ranking with increasing E_Homo_ value is as follows:Compound **5d** > Compound **5f**> Compound **5e** > Compound **5h** > Compound **5a** > Compound **5c** > Compound **5b** > Compound **5g**

The second computational parameter is E_Lumo_. The low value of a compound predicts that it can easily accept electrons from a molecule, thus resulting in the increase of biological activity with decreasing E_Lumo_ values. Thus, according to Table 3, the increase in biological activity of compounds is as follows:Compound**5a** > Compound **5b** > Compound **5c** > Compound **5g** > Compound **5d** > Compound **5f** > Compound **5e** > Compound **5h**

The third parameter is the energy gap ΔE between HOMO and LUMO. A smaller energy gap is associated with a more reactive molecule that is kinetically less stable. Thus, biological activity of synthesized compounds increases with decrease in the energy gap. The order of ranking should be [38,40,41]:Compound **5f** > Compound **5e** > Compound **5h** > Compound **5d** > Compound **5a** > Compound **5c** ≈ Compound **5g** > Compound **5b**

Another property associated with FMOs is chemical softness and hardness. According to HSAB approximation, hard molecules have large energy gaps and soft molecules have smaller energy gaps. As the biological system consists of enzymes and cells which are soft, soft molecules tend to coordinate more easily. Thus, increasing biological activity of compounds according to the global softness criterion is as follows [38]:Compound **5h** > Compound **5b** > Compound **5c** ≈ Compound **5g** > Compound **5a** > Compound **5d** > Compound **5e** > Compound **5f**

#### 2.3.2. Global Reactivity Descriptors

Electronegativity (X), electrophilicity index (ω), Dipole Moment (µ), Electronic energy (E), Ionization Potential (I), Chemical Potential (CP), Nucleophilicity Index (N) and electronic charges (ΔN_max_) are reactivity parameters. Table 3, E shows the calculated values.

A low value of electronegativity (X) or higher value of chemical potential (CP) refers to the electron delocalization. This means that a molecule can easily coordinate with a biological system by forming bonds. According to this criterion, increasing biological ranking should be as follows:Compound **5a** > Compound **5c** ≈ Compound **5d >** Compound **5b** > Compound **5f** ≈ Compound **5g** > Compound **5e** > Compound **5h**

The next parameters are electrophilicity (ω) and nucleophilicity (N) indexes. An increase in biological activity is associated with an increasing value of nucleophilicity (N) index and decreasing value of electrophilicity (ω), therefore these molecules showed best binding affinities in case of alpha amylase protein interaction with these molecules for screening anti-diabetic activity virtually. Owing to this property, as **5a** has the highest nucleophilicity and lowest electrophilicity, it therefore showed seven hydrogen bonding interactions with highest MolDock compared to other compounds in case of protein ligand interaction with alpha amylase binding site.

The order of ranking should be:Compound **5a** > Compound **5b** ≈ Compound **5c** > Compound **5g** > Compound **5d** > Compound **5f** > Compound **5e** > Compound **5h**

ΔN_max_ is the charge of compounds. If ΔN_max_ increases then biological activities also increase. A high value of dipole moment of compound **5f** is an indication of fine charge distribution and bond distance adjusted well. This explains that the molecule showing the process of oxidation reflects the best conductivity [38,42,43,44].

#### 2.3.3. Molecular Electrostatic Potential (MEP)

Molecular electrostatic potential (MEP) helps us in assisting the inter and intra molecular interactions like hydrogen bonding, nucleophilic and electrophilic interactions with incoming molecules, and helps in recognition of biological interactions like molecular docking, drug protein interactions, etc. The MEP of the compounds **5a**–**h** is shown in Figure 15 based on SCF energy. MEP shows the response and behavior of molecules toward the binding sites in biological system. It acts as a visual scheme for accessing the polarity of molecule. The positive (blue) part of MEP shows sites of attack of nucleophiles while negative (red and yellow) areas show the sites of attack of electrophilic molecules. Thus, all the synthesized molecules have blue, or positive, regions around all hydrogen atoms and specifically more blue color around the hydrogen atoms linked to nitrogen moiety indicating high electron density, and red or yellow regions around the oxygen atoms of carbonyl group present between two nitrogen moiety of pyrimidine ring. The neutral portion is of benzene molecules, having a green color in MEP. Localization of blue regions over hydrogen atoms of benzene rings and localization of yellow or red color over oxygen atom of pyrimidine moiety is responsible for the best radical scavenging activity [39,41,45].

## 3. Methodology

### 3.1. Chemistry

The materials (chemicals and solvents) used for synthesis were 95% pure of the brand Alfa Aesar Gemany and Daejung Korea, purchased from local vendor Musa ji Adam & Sons in Faisalabad, Pakistan. These were used as such, with no need for further purification. Aluminum pre-coated plates (Silica gel 60 F254 Merck KGaA, Darmstadt, Germany) were used for analytical thin layer chromatography with ethyl acetate/n-hexane (7:3) as eluent. The melting point of compounds was measured by Stuart melting point apparatus SP10. ^1^H-NMR spectra were recorded on Bruker 300 MHz NMR spectrometer, and ^13^C-NMR on Bruker 25 and 175 MHz using DMSO-*d*_6_ as solvent and TMS as internal standard.

### 3.2. General Procedure of Synthesis

For the synthesis of required compounds, ecofriendly, one-pot multi-component methodology was used with some modifications (Figure 16). In this method, a mixture of multi-component reagents, i.e., ethyl acetoacetate, urea/thiourea and aromatic benzaldehyde, was stirred together in a round bottom flask for about 10 min, adding a catalytic amount of CuCl_2_·H_2_O so that aromaticity could be achieved. Then the mixture obtained was left overnight for obtaining best results. After this the mixture was washed with cold water to remove excess copper salt, dried and re-crystallized with hot solvents to achieve pure products (**5a**–**h**). Progression of reaction was checked by TLC [46,47].

#### 3.2.1. Synthesis of Ethyl 4-(Fluorophenyl)-6-methyl-2oxo-1,2,3,4-tetrahydropyrimidine-5-carboxylate (**5a**–**f**)

The appropriate amounts, i.e., 0.2 M (6.2 g) of fluoro-benzaldehyde, urea (3 g), thiourea (3.8 g) and ethyl acetoacetate (6.5 g), were mixed by adding a catalytic amount of CuCl_2_·2H_2_O by grinding for 7–0 min. After adding a few drops of HCl, they were again mixed for about 10 min. The mixture was then allowed to stayed overnight. Products obtained were purified by dissolving in methanol, by slightly heating the solution in a water bath. The solution was filtered off and allowed to re-crystallize. The fate of the performed reaction was determined with TLC.

##### Spectral Data of 4-(Fluorophenyl)-6-methyl-2oxo-1,2,3,4-tetrahydropyrimidine-5-carboxylate

4-(2-fluorophenyl)-6-methyl-2oxo-1,2,3,4-tetrahydropyrimidine-5-carboxylate (**5a**): White shiny crystalline solid, soluble in DMSO, M.P = 230–233 °C; ^1^H-NMR (DMSO, 300 MHz): δ 9.26 (1H, *s*, NH), 7.70 (1H, *s*, NH), 7.30 (1H, *m*, H-4′), 7.26 (1-H, *m*, H-5′), 7.11 (1H, *m*, H-3′), 7.17 (1H, *m*, H-6′), 5.44 (1H, *s*, H-4), 3.94 (1H, *q*, *J* = 6.9 Hz, CH_2_), 2.26 (1H, *s*, CH_3_), 1.05 (1H, *t*, CH_3_); ^13^C-NMR (DMSO, 25 MHz): δ 165.3 (C-1″), 100.3 (C-5), 60.2 (C-3″), 14.4 (C-4″), 17.6 (CH_3_), 146.1 (C-6), 54.07 (C-4), 152.0 (C-2), 143.9 (C-1′), 122.3 (C-2′), 161.3 (C-3′), 112.5 (C-4′), 130.2 (C-5′), 113.8 (C-6′); EIMS: *m*/*z* 278.12 (cacld. For C_14_H_15_FN_2_O_3_).

ethyl 4-(3-fluorophenyl)-6-methyl-2oxo-1,2,3,4-tetrahydropyrimidine-5-carboxylate (**5b**): White shiny crystalline solid, soluble in DMSO, M.P = 210–213 °C; ^1^H-NMR(DMSO, 300 MHz): δ 9.27 (1H, *s*, NH), 7.81 (1H, *s*, NH), 7.40 (1H, *dd*, H-5′), 7.36 (1H, *d*, H-4′), 7.28 (1-H, *d*, H-6′), 6.98 (1H, *s*, H-2′), 5.17 (1H, *s*, H-4), 4.03 (1H, *q*, *J* = 9 Hz, CH_2_), 2.50 (1H, *s*, CH_3_), 1.10 (1H, *t*, CH_3_); ^13^C-NMR (DMSO, 25 MHz): δ 165.69 (C-1″), 152.41 (C-2), 149.19 (C-6), 144.21 (C-1′), 132.26 (C-3′), 129.35 (C-5′), 128.88 (C-4′), 128.66 (C-2′), 128.66 (C-6′), 99.29 (C-5), 59.77 (3″-CH_2_), 53.85 (C-4), 18.25 (CH_3_), 14.51 (4″-CH_3_); EIMS: *m*/*z* 278.12 (cacld. For C_14_H_15_FN_2_O_3_).

ethyl 4-(4-fluorophenyl)-6-methyl-2oxo-1,2,3,4-tetrahydropyrimidine-5-carboxylate (**5c**): White shiny crystalline solid, soluble in DMSO, M.P = 182–184 °C; ^1^H-NMR (DMSO, 300 MHz): δ 9.23 (1H, *s*, NH), 7.76 (1H, *s*, NH), 7.28 (1H, *d*, H-3′), 7.26 (1H, *d*, H-5′), 7.15 (1-H, *d*, H-6′), 7.14 (1H, *d*, H-2′), 5.15 (1H, *s*, H-4), 4.01 (1H, *q*, *J* = 6.9 Hz, CH_2_), 2.50 (1H, *s*, CH_3_), 1.10 (1H, *t*, CH_3_); ^13^C-NMR (DMSO, 25 MHz): δ 165.70 (C-1″), 152.48 (C-2), 148.98 (C-6), 141.60 (C-1′), 128.66 (C-3′), 128.77 (C-5′), 115.70 (C-6′), 115.42 (C-2′), 160.17 (C-4′), 99.58 (C-5), 59.67 (3″-CH_2_), 53.81 (C-4), 18.23 (CH_3_), 14.49 (4″-CH_3_); EIMS: *m*/*z* 278.12 (cacld. For C_14_H_15_FN_2_O_3_).

ethyl 4-(2-fluorophenyl)-6-methyl-2-thioxo-1,2,3,4-tetrahydropyrimidine-5-carboxylate (**5d**): White shiny crystalline solid, soluble in DMSO, Decomposing temperature = 216–220 °C; ^1^H-NMR (DMSO, 300 MHz): δ 10.56 (1H, *s*, NH), 9.97 (1H, *s*, NH), 7.25 (1H, *m*, H-5′), 7.22 (1-H, *m*, H-4′), 7.19 (1H, *m*, H-6′), 7.17 (1H, *m*, H-3′), 5.55 (1H, *s*, H-4), 3.98 (1H, *q*, *J* = 6 Hz, CH_2_), 2.33 (1H, *s*, CH_3_), 1.05 (1H, *t*, CH_3_); ^13^C-NMR (DMSO, 25 MHz): δ 165.02 (C-1″), 173.06 (C-2), 161.42 (C-6), 129.79 (C-1′), 115.24 (C-3′), 130.34 (C-4′), 125.24 (C-5′), 158.14 (C-2′), 130.52 (C-6′), 100.29 (C-5), 60.13 (C-4), 49.34 (C-3″); 17.56 (CH_3_), 14.24 (C-4″); EIMS: *m*/*z* 294.02 (cacld. For C_14_H_15_FN_2_O_2_S).

ethyl 4-(3-fluorophenyl)-6-methyl-2-thioxo-1,2,3,4-tetrahydropyrimidine-5-carboxylate (**5e**): White shiny crystalline solid, soluble in DMSO, M.P = 202–206 °C; ^1^H-NMR (DMSO, 300 MHz): δ 10.43 (1H, *s*, NH), 9.75 (1H, *s*, NH), 7.44 (1H, *dd*, H-5′), 7.08 (1H, *d*, H-4′), 7.15 (1-H, *d*, H-6′), 6.97 (1H, *s*, H-2′), 5.21 (1H, *s*, H-4), 4.05 (1H, *q*, *J* = 6.3 Hz, CH_2_), 2.50 (1H, *s*, CH_3_), 1.12 (1H, *t*, CH_3_); ^13^C-NMR (DMSO, 25 MHz): δ 165.43 (C-1″), 174.61 (C-2), 160.97 (C-6), 146.51 (C-1′), 164.21 (C-3′), 131.27 (C-5′), 113.73 (C-4′), 113.44 (C-2′), 122.87 (C-6′), 100.74 (C-5), 60.18 (3″-CH_2_), 53.99 (C-4), 17.67 (CH_3_), 14.46 (4″-CH_3_); EIMS: *m*/*z* 294.02 (cacld. For C_14_H_15_FN_2_O_2_S).

ethyl 4-(4-fluorophenyl)-6-methyl-2thioxo-1,2,3,4-tetrahydropyrimidine-5-carboxylate (**5f**): White shiny crystalline solid, soluble in DMSO, M.P = 233–235 °C;^1^H-NMR (DMSO, 300 MHz): δ 10.43 (1H, *s*, NH), 9.77 (1H, *s,* NH), 7.27 (1H, *d,* H-3′), 7.24 (1H, *d*, H-5′), 7.18 (1-H, *d*, H-6′), 7.15 (1H, *d,* H-2′), 5.20 (1H, *s*, H-4), 4.04 (1H, *q,* J = 6.6 Hz, CH_2_), 2.30 (1H, *s*, CH_3_), 1.11 (1H, *t*, CH_3_); ^13^C-NMR (DMSO, 25 MHz): δ 165.49 (C-1″), 174.57 (C-2), 163.60 (C-6), 140.23 (C-1′), 115.68 (C-3′), 115.68 (C-5′), 128.97 (C-6′), 128.86 (C-2′), 160.38 (C-4′), 101.95 (C-5), 60.09 (3″-CH_2_), 53.83 (C-4), 17.63 (CH_3_), 14.46 (4″-CH_3_); EIMS: *m*/*z* 294.02 (cacld. For C_14_H_15_FN_2_O_2_S).

##### Synthesis of Ethyl 4-(Chlorophenyl)-6-methyl-2oxo-1,2,3,4-tetrahydropyrimidine-5-carboxylate (**5****g**,**h**)

The appropriate amounts, i.e., 0.1 M (3.5 g) of chloro-benzaldehyde, urea (3 g), thiourea (3.8 g) and ethyl acetoacetate (6.5 g), were mixed by adding a catalytic amount of CuCl_2_·2H_2_O by grinding for 7–10 min. After adding a few drops of HCl, they were again mixed for about 10 min. The mixture was then allowed to stayed overnight. Products obtained were purified by dissolving in methanol, by slightly heating the solution in a water bath. The solution was filtered off and allowed to re-crystallize. The fate of the performed reaction was determined with TLC.

ethyl 4-(3-chlorophenyl)-6-methyl-2oxo-1,2,3,4-tetrahydropyrimidine-5-carboxylate (**5g**): White shiny crystalline solid, soluble in DMSO, M.P = 215–220 °C; ^1^H-NMR (DMSO, 300 MHz): δ 9.34 (1H, *s*, NH), 7.86 (1H, *s*, NH), 7.37 (1H, *s*, H-2′), 7.31 (1H, *d*, H-4′), 7.31 (1-H, *dd*, H-5′), 7.24 (1H, *d*, H-6′), 5.21 (1H, *s*, H-4), 4.01 (1H, *q*, *J* = 6 Hz, CH_2_), 2.29 (1H, *s*, CH_3_), 1.10 (1H, *t*, CH_3_); ^13^C-NMR (DMSO, 175 MHz): δ 165.6 (C-1″), 152.5 (C-2), 149.3 (C-6), 147.6 (C-1′), 133.4 (C-3′), 130.0 (C-5′), 127.6 (C-4′), 126.7 (C-2′), 125.3 (C-6′), 99.1 (C-5), 59.7 (3″-CH_2_), 54.1 (C-4), 18.2 (CH_3_), 14.4 (4″-CH_3_); EIMS: *m*/*z* 294.72 (cacld. For C_14_H_15_ClN_2_O_3_).

ethyl 4-(3-chlorophenyl)-6-methyl-2thioxo-1,2,3,4-tetrahydropyrimidine-5-carboxylate (**5h**): White shiny crystalline solid, soluble in DMSO, M.P = 182–188 °C; ^1^H-NMR (DMSO, 300 MHz): δ 10.41 (1H, *s*, NH), 9.71 (1H, *s*, NH), 7.73 (1H, *s*, H-2′), 7.39 (1H, *d*, H-4′), 7.36 (1-H, *dd*, H-5′), 7.16 (1H, *d*, H-6′), 5.19 (1H, *s*, H-4), 4.05 (1H, *q*, *J* = 6 Hz, CH_2_), 2.29 (1H, *s*, CH_3_), 1.11 (1H, *t*, CH_3_); ^13^C-NMR (DMSO, 175 MHz): δ 174.5 (C-2), 165.3 (C-1″), 146.1 (C-6), 145.9 (C-1′), 133.5 (C-3′), 131.1 (C-5′), 128.1 (C-4′), 126.8 (C-2′), 125.5(C-6′), 100.6 (C-5), 60.2 (3″-CH_2_),54.0 (C-4), 17.6 (CH_3_), 14.4 (4″-CH_3_); EIMS: *m*/*z* 310.0 (cacld. For C_14_H_15_ClN_2_O_2_S).

### 3.3. Biological Assays:

#### 3.3.1. Antioxidant (Free Radical Scavenging Activity) Using DPPH

The potential of synthesized compounds to scavenge the free radicals were assessed by using diphenylpicrylhydrazyl (DPPH) as free radical [48].

##### Preparation of Stock Solution

In order to perform anti-oxidant activity, stock solutions of compounds **5a**–**h** having concentration 200 mM (0.022 g) were prepared. These stock solutions were then subjected to prepare their dilutions up to 250 µg, 200 µg, 150 µg, 100 µg, 50 µg and 25 µg respectively. DPPH solution of 4% was prepared by mixing 0.004 g of DPPH in 100 ml methanol (commercial grade). Ascorbic acid was used as control and its solution was prepared by mixing 0.0017612 g into 100 ml of water having 100 µg concentration.

##### Protocol of Free Radical Scavenging Activity

In test tubes 1 mL of sample from each diluted solution and 2 mL of DPPH solution was added. In the same way, 1 mL of the ascorbic acid was added along with 2 ml of DPPH as control and allowed to stand overnight. The next day, using Hitachi U-2900 spectrophotometer, readings at 517 nm were taken in triplicate and their inhibitory concentration calculated using formula [49].
$$A=\frac{Acontrol-Asample}{Acontrol}\;\times 100$$

#### 3.3.2. Anti-Diabetic Activity

Synthesized compounds **5a**–**h** were subjected to in vitro anti-diabetic activity using starch as substrate and alpha amylase as enzyme. Acarbose was used as standard drug [50].

##### Protocol of Anti-Diabetic Activity

In a test tube 0.5 mL of compound of each specific dilution and 0.5 mL of enzyme were added and kept in incubator at room temperature for 10 min, as done by our previous method [51]. Readings were taken in triplicate at 540 nm and percentage for the inhibition activity of target molecule was measured by the formula [52].
$$A=\frac{Acontrol-Asample}{Acontrol}\;\times 100$$

#### 3.3.3. Anti-Cancer Activity

The human HepG2 cells were cultured in Dulbecco’s Modified Eagle’s Medium (DMEM) supplemented with 10% fatal bovine serum (FBS), 100 unit’s/mL penicillin and 100 μg/mL streptomycin, and maintained at 37 °C with 5% CO_2_ in humidified atmosphere. Cells were treated with extracts/compound dissolved in DMSO with a final DMSO concentration of 0.05%. DMSO treated cells were used as control in all the experiments [53].

##### Determination of Cell Viability

Cell viability was determined by MTT assay as described by us previously. Briefly, HepG2 cells were treated with different concentrations of compounds for 48 h. Following treatment, the MTT reagent was added (500 μg/mL) and cells were further incubated at 37 °C for 4 h. Subsequently, 150 μL DMSO was added to dissolve formazan crystals and absorbance was measured at 490 nm in a microplate reader (Thermo Scientific, Waltham, MA, USA). The percentage of cell viability was calculated [54].

### 3.4. Molecular Docking Studies

#### 3.4.1. Ligand and Protein Preparation

All the synthesized compounds were drawn into 2D conformers by using ChemDraw Professional (Version 19.1.0.8, Perkin Elmer, Waltham, MA, USA), then these structures were imported to chem3D Ultra (Version 19.1.0.8, Perkin Elmer, Waltham, MA, USA). All the energy minimizations and other structural conformations, such as bond order, assigning missing bonds, explicit hydrogens and flexible torsional if absent in synthesized compounds, were optimized by using MM2 force field-Steepest Descent Algorithm in chem3D [27,55]. The crystal structure of target protein was downloaded from Protein Data Bank (PDB) at the Research Collaboratory for Structural Bioinformatics (RCSB, http://www.rcsb.org, accessed on 13–17 January 2021). PDB ID for anti-cancer activity is human serum albumin having ID (3B9L) with resolution of 2.60 Å. For antidiabetic activity, the crystal structure of human glucokinase having PDB ID (1V4S) with a resolution of 2.3 Å was used as target receptor site. In case of anti-oxidant activity, quercetin complex kinase having PDB ID (2HCK) with a resolution of 3.00 Å was used as target crystal structure. All these crystal structures were downloaded as PDB Format (gz) and then, after importing into MVD, their protein preparation was carried out before docking [22,56,57,58,59].

#### 3.4.2. Molecular Docking

Molecular docking studies based on synthesized compounds 5a-c was conducted by using Molegro Virtual Docker (MVD 2013 6.0.1, Molegro ApS, Aarhus, Denmark) and Discovery Studio Visualizer 2020. As studies revealed that MVD showed the best binding affinity in the form of Mol dock score as compared to other docking software like Surflex, Glide, Autodock Vina and FlexX [22,60]. Pose comprising top docked conformation was examined by using MolDock Score [GRID] algorithm, number of runs taken as 10, maximum iterations were 1500, population size of 50, with energy threshold of 100 having grid resolution 0.30 Å [23,24,25,26,27,58,61,62]. The MolDock score, length and number of hydrogen bonds involved along with residues involved in hydrogen bonding are listed for all the biological activities.

### 3.5. Density Functional Theory Studies of Target Molecules in Scope of QSAR

Density functional language is a universal approach for defining the molecular structures of organic compounds and their activity relationships by making use of quantum-chemical descriptors and geometrical parameters [63]. The main objective is the correlation of biological activities of compounds with molecular descriptors given by DFT calculations [64]. All computational calculations of compounds were carried through Gaussian 09W program supported by the Gauss View 6.0.16 interface [65]. The molecules are optimized and its parameters calculated operating hybrid type B3LYP utilizing 321-G basis set with DFT exchange, which provide HOMO-LUMO geometries, net charge, energy gap, dipole moment and other computational descriptors [40].

## 4. Conclusions

Eight new compounds were synthesized in one-pot multi-component strategy by adopting solvent free methodology. All the compounds **5a**–**h** was evaluated for their biological potential against diabetes, radical scavenge and cancer cell lines. After combinatorial in vitro and in silico analysis, compound **5c** was pharmacologically active for anti-oxidant activity, **5e** and **5c** for cytotoxicity and **5g** for combating diabetes. These potential leads had the best biological potential, and their biological activity can be enhanced by further futural derivatization, or by making their structural analogs. Hence, overall, the synthesized target molecules were proved to be potential therapeutic agents against different classes of inhibitor, thus opening new research interests regarding pyrimidine derivatives that may lead to evolution in medicinal systems and in the field of synthesis. The conclusion is that the synthetic pathway chosen here can be of primary interest for research communities and pharmacists, and the novel compounds synthesized here can be used in future as effective drugs due to their cell specificity, safety, inexpensiveness, sustainability, enhanced activity, eco-friendly nature and less time consuming and cost-effective synthetic method.

## Figures and Tables

**Figure 1 molecules-26-04424-f001:**
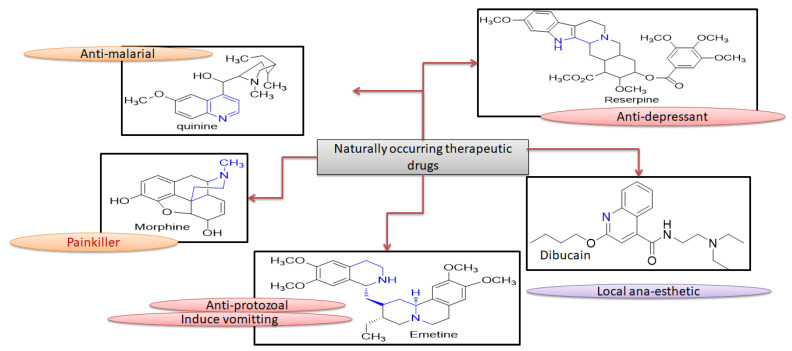
Naturally occurring therapeutic drugs with their potentials.

**Figure 2 molecules-26-04424-f002:**
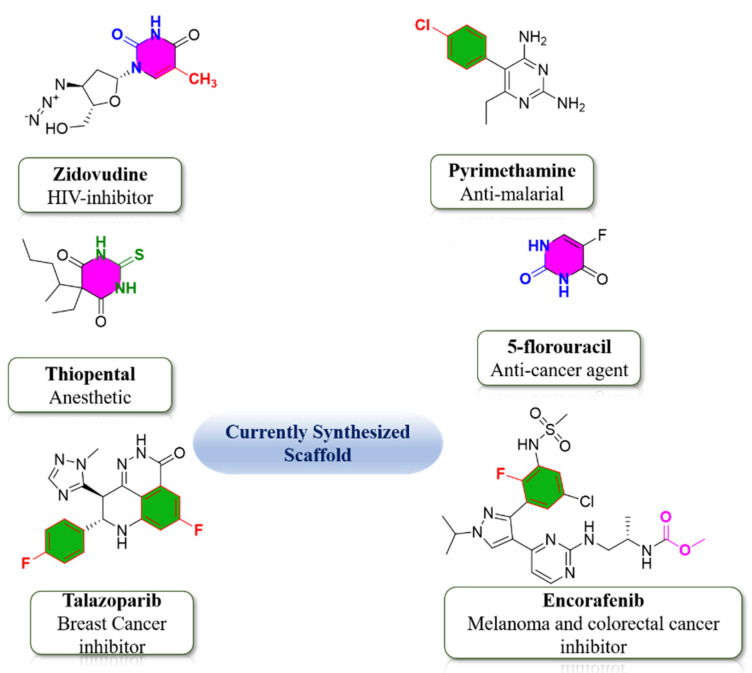
Biologically active functionality group-based currently synthesized Pyrimidine-scaffold.

**Figure 3 molecules-26-04424-f003:**
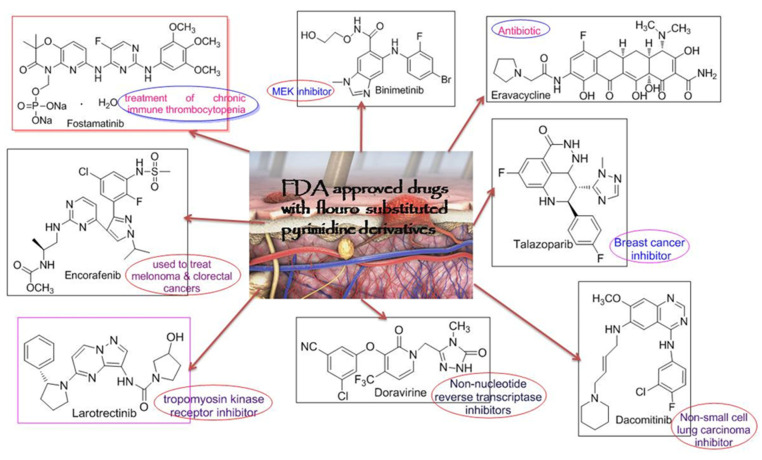
FDA approved drugs with pyrimidine skeleton having fluoro-substitution.

**Figure 4 molecules-26-04424-f004:**
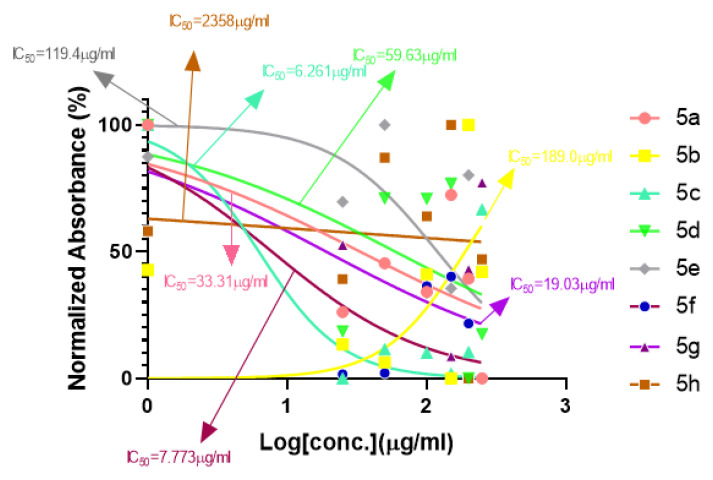
Antioxidant activities of compounds **5a**–**h** with IC_50_ (ascorbic acid as standard).

**Figure 5 molecules-26-04424-f005:**
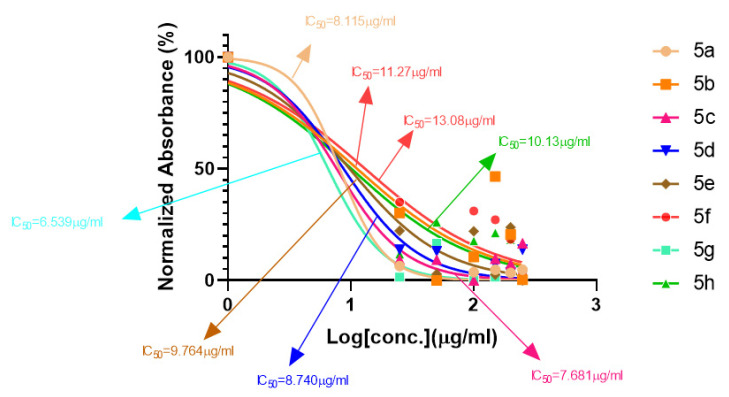
Graphical representation of anti-diabetic activity of **5a**–**h:** Acarbose as standard.

**Figure 6 molecules-26-04424-f006:**
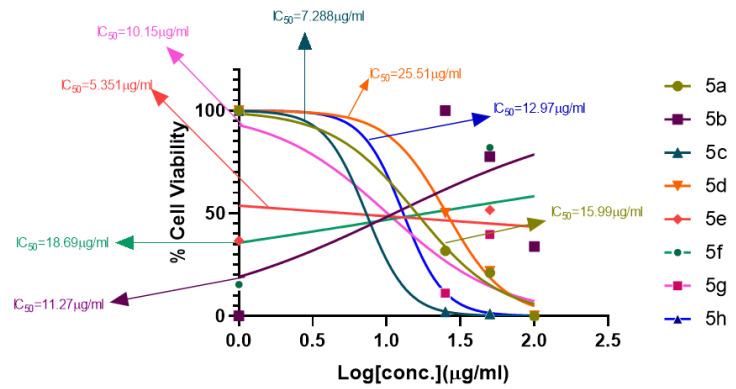
MTT cytotoxicity assay IC_50_ values interpretation.

**Figure 7 molecules-26-04424-f007:**
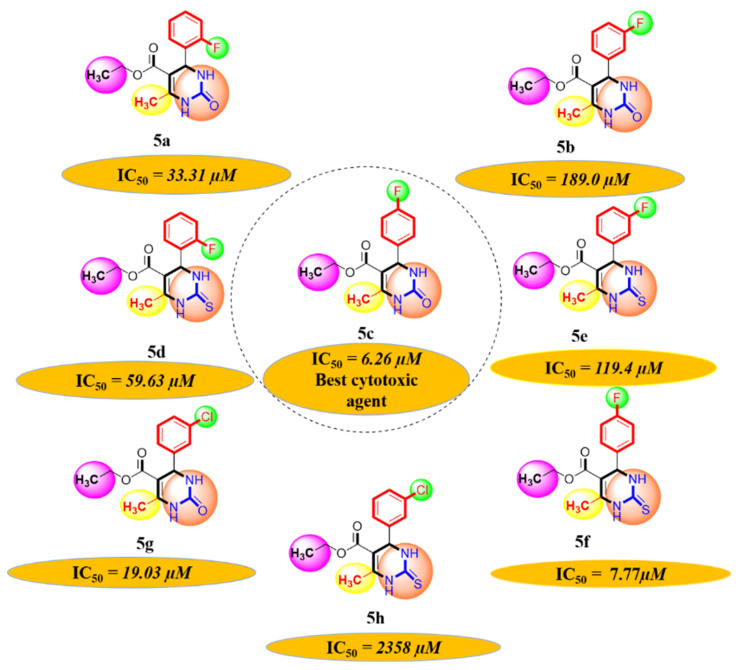
Diagrammatic relationship of target molecules (**5a–h**) for their structure-activity relationship as anti-oxidant agents.

**Figure 8 molecules-26-04424-f008:**
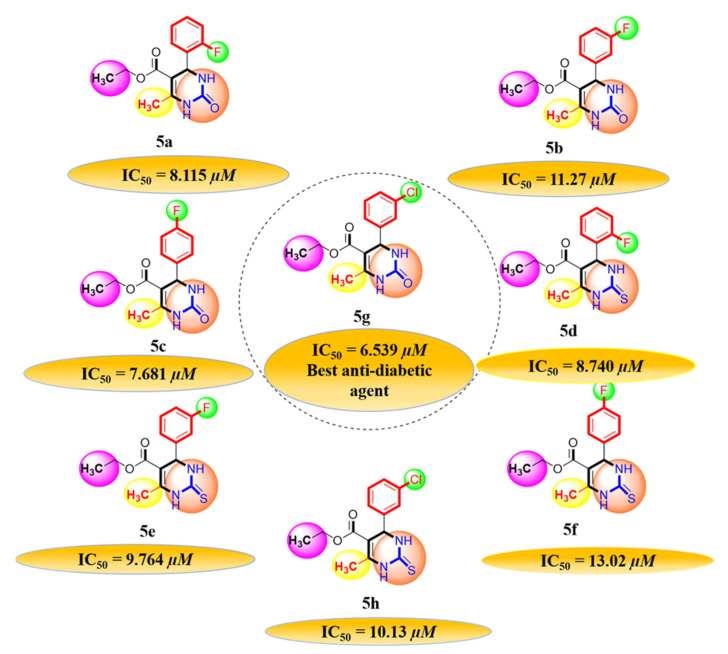
Diagrammatic relationship of target molecules (**5a**–**h**) for their structure-activity relationship as anti-diabetic agents.

**Figure 9 molecules-26-04424-f009:**
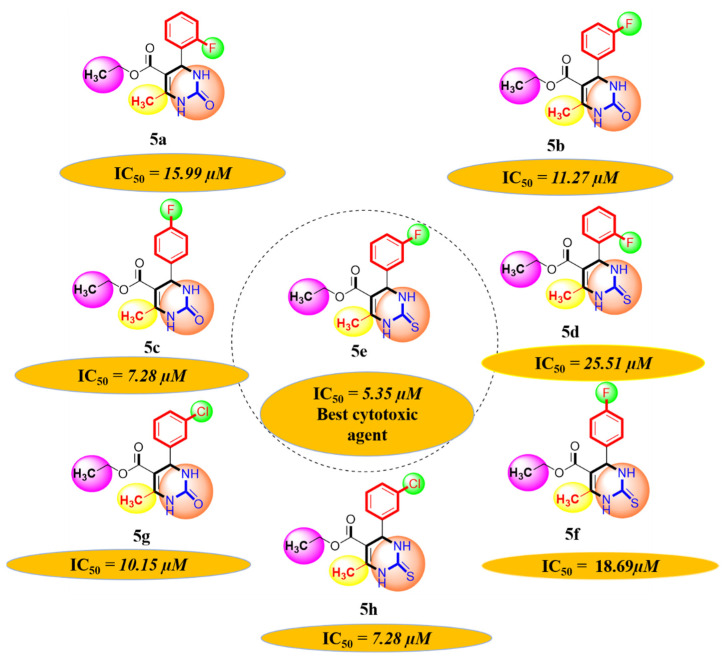
Diagrammatic relationship of target molecules (**5a**–**h**) for their structure-activity relationship as anti-cancer agents.

**Figure 10 molecules-26-04424-f010:**
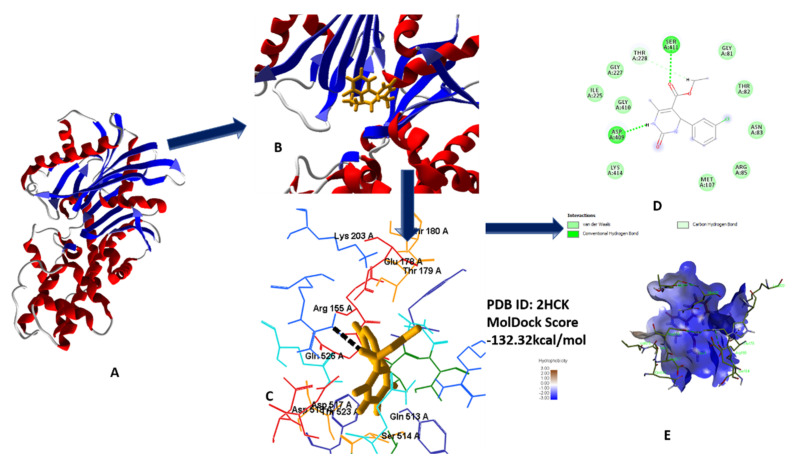
(**A**) Secondary structure of 2HCK; (**B**) secondary structure along with lead compound **5c**; (**C**) 3D interaction with protein; (**D**) 2D diagram; (**E**) hydrophobic interaction of **5c** with 2HCK.

**Figure 11 molecules-26-04424-f011:**
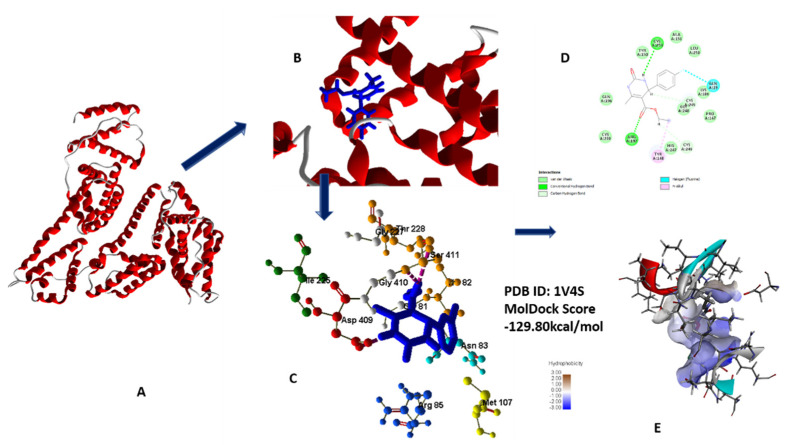
(**A**) Secondary structure of 1V4S; (**B**) secondary structure along with lead compound **5g**; (**C**) 3D interaction with protein; (**D**) 2D diagram; (**E**) hydrophobic interaction of **5g** with 1V4S.

**Figure 12 molecules-26-04424-f012:**
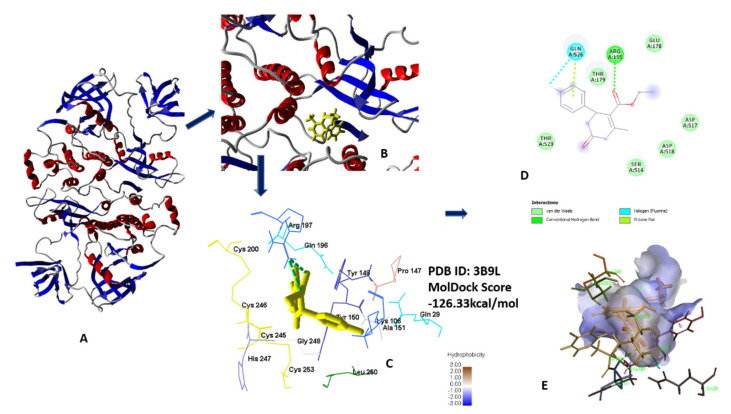
(**A**) Secondary Structure of 3B9L; (**B**) secondary structure along with lead compound **5c**; (**C**) 3D interaction with protein; (**D**) 2D diagram; (**E**) hydrophobic interaction of **5c** with 3B9L.

**Figure 13 molecules-26-04424-f013:**
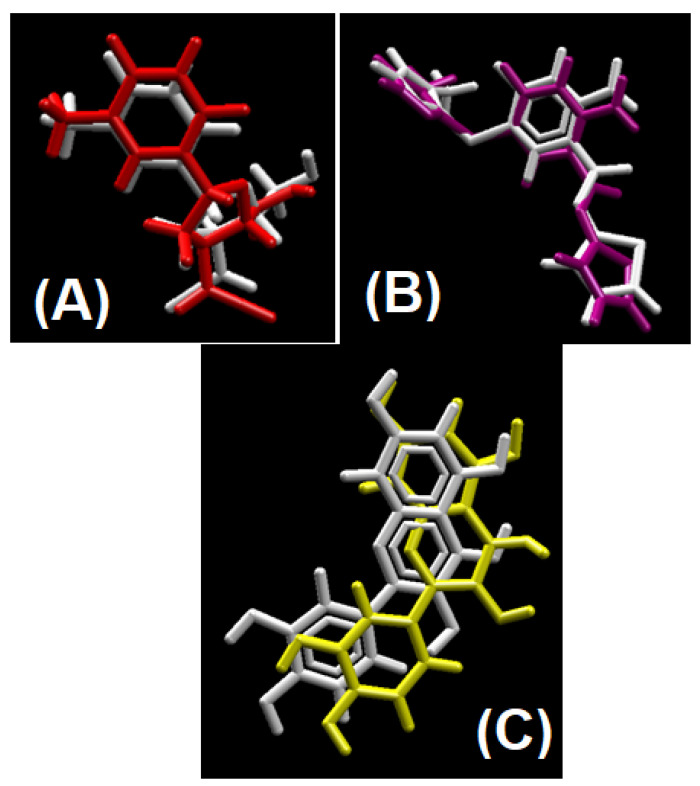
Validation of docking protocol by superimposing native ligand to redocked ligand to their respective co-crystal structure. (**A**) Redocked ligand (red) superimpose to native ligand (white) for anti-cancer docking protocol validation; (**B**) redocked ligand (purple) superimpose to native ligand (white) for anti-diabetic docking protocol validation; (**C**) redocked ligand (yellow) superimpose to native ligand (white) for antioxidant docking protocol validation.

**Figure 14 molecules-26-04424-f014:**
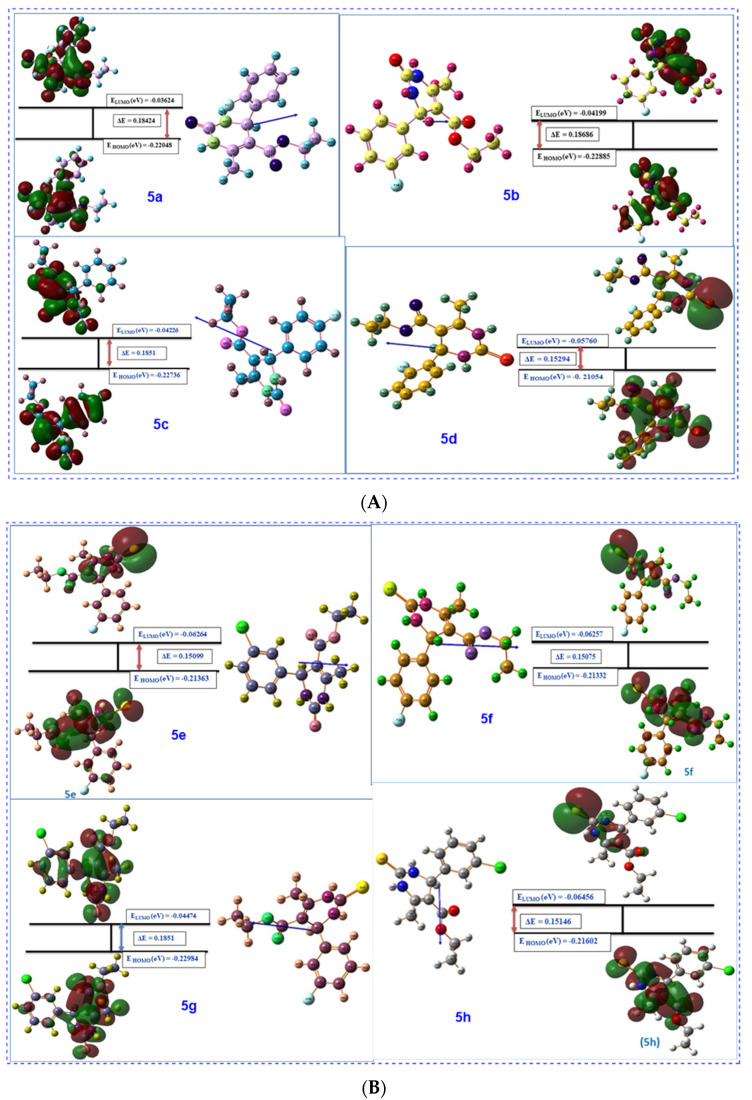
(**A**) Contour diagrams of FMOs comprising the highest occupied molecular orbitals (HOMO) and the lowest unoccupied molecular orbitals (LUMO), along with energy gap (ΔE), optimized geometry of structures with numbering system and the vector of the dipole moment for compounds **5a**–**d**. (**B**) Contour diagrams of FMOs comprising the highest occupied molecular orbitals (HOMO) and the lowest unoccupied molecular orbitals (LUMO), along with energy gap (ΔE), optimized geometry of structures along with numbering system and the vector of the dipole moment for compounds **5e**–**h**.

**Figure 15 molecules-26-04424-f015:**
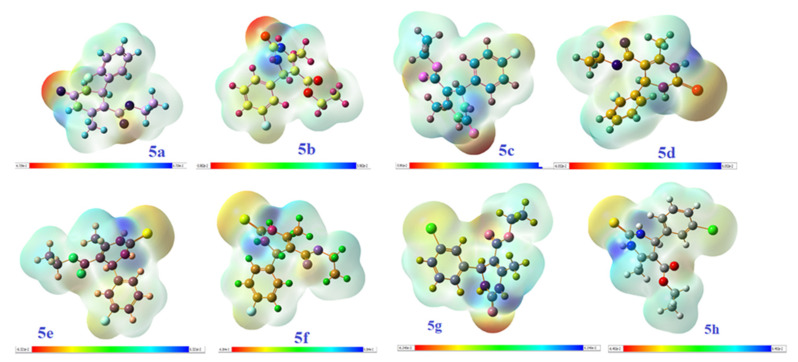
MEP of the compounds **5a**–**h** based on SCF energy.

**Figure 16 molecules-26-04424-f016:**
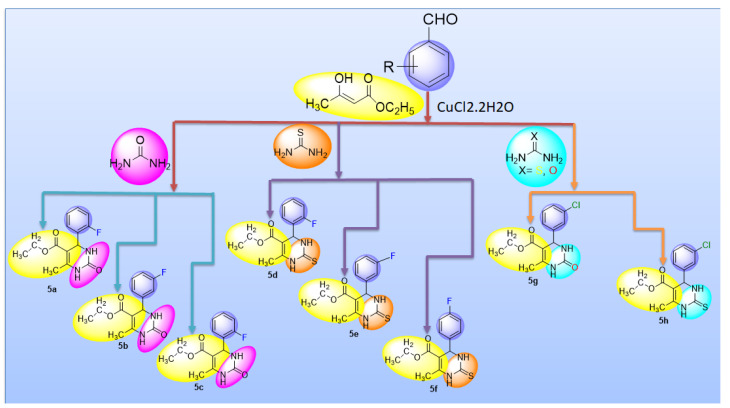
General equation for synthesis of compounds (**5a**–**h**).

**Table 1 molecules-26-04424-t001:**
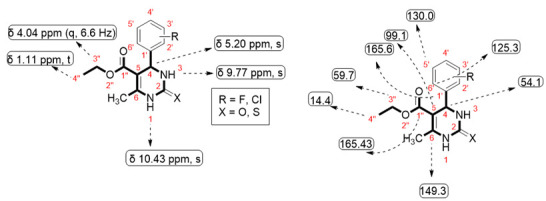
^13^C-NMR (DMSO, 25 MHz) δ values.

Carbon Atom Number	Functional Group R = F, Cl	Chemical Shift δ (ppm)
2-C′	F	132.26
2-C′	H	113.44
3-C′	F	132.26
3-C′	H	115.24
4-C′	F	160.17
4-C′	H	128.88
5-C′	H	125.24
6-C′	H	130.52
3-C′	Cl	133.4

**Table 2 molecules-26-04424-t002:** ^13^C-NMR (DMSO, 25 MHz) δ values.

Carbon Number	Functional Group X = O, S	Chemical Shift δ (ppm)
2-C	O	152.48
2-C	S	174.57

**Table 3 molecules-26-04424-t003:** Quantum and geometrical-based computational parameters based on DFT calculations having 321-G basis set to facilitate SAR studies of compounds, where EHOMO (eV), ELUMO (eV), Energy Gap “∆E = ELUMO-EHOMO (eV)”, Dipole Moment “μ(Debye)”, Global Hardness “η (eV)”.

Molecular Structure Activity Relationship	Quantum Chemical Parameters												
	E_HOMO_ (eV)	E_LUMO_ (eV)	ΔE (eV)	µ (Debye)	η (eV)	S (eV)	χ (eV)	CP	N	Ω	I (eV)	E (Hatree)	ΔN_max_
**5a**	−0.22048	−0.03624	0.18424	2.996	0.0921	5.427	0.128	−0.128	11.183	0.08942	0.22048	−972.22	1.389
**5b**	−0.22885	−0.04199	0.18686	1.070	0.09343	5.351	0.135	−0.135	10.189	0.09814	0.22885	−972.22	1.444
**5c**	−0.22736	−0.04226	0.1851	2.384	0.09255	5.402	0.134	−0.134	10.185	0.09818	0.22736	−972.22	1.447
**5d**	−0.21054	−0.05760	0.15294	4.441706	0.07674	6.538	0.134	−0.134	8.510	0.1175	0.21054	−1293.64	1.753
**5e**	−0.21363	−0.06264	0.15099	4.622320	0.07549	6.623	0.138	−0.134	7.917	0.1263	0.21363	−1293.64	1.829
**5f**	−0.21332	−0.06257	0.15075	5.548142	0.07537	6.633	0.137	−0.137	7.930	0.1261	0.21332	−1293.64	1.829
**5g**	−0.22984	−0.04474	0.1851	3.909035	0.09255	5.402	0.137	−0.137	9.842	0.1016	0.22984	−1330.91	1.480
**5h**	−0.21602	−0.06456	0.15146	4.660382	0.07573	6.602	0.140	−0.140	7.710	0.129	0.21602	−1652.33	1.851

## Data Availability

Not applicable.

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
