# Peer review of "Structure-Based Designing, Solvent Less Synthesis of 1,2,3,4-Tetrahydropyrimidine-5-carboxylate Derivatives: A Combined In Vitro and In Silico Screening Approach"

_molecules, 2021, doi:10.3390/molecules26154424_

Round 1
Reviewer 1 Report
Authors did choose their compounds basing on the already known physiologic acyivity of compounds bearing pyrimidine scaffold. It is o.k. to me, however, the rationale behind the choice of specific activities are not clear. This is also seen in Abstract where the goal of the paper is chaotically presented.
This paper is too spacious and too many aspects of potential activity of the synthesized compounds are presented in a very detailed form. Because it is also badly written it is not suitable for publication in present form.
Specific comments are as follows (in order of appearance):
1./ highlights are not informative (I am not sure if they are already required);
2./ procaine is not a heterocycle (page 2);
3./ there is no need to write some names of the compounds in capital letters;
4./ Statement "This technique is efficient for the synthesis of lead compounds by solid mixing of reactants in an agate mortarand pestle without using any kind of solvent" tries to present the adantages of synthesis in sold phase but is awkward; there is a number of such statements in the text;
5./ urea and its analogs are not secondary amines (page 4 line 110);
6./ when providing the producers their location should be indentified;
7./ 10 min not mins (page 4 line 111);
8./ Statement: "The aqua washed off formulated compound (5a) and solubility was verified in an appropriate solvent after drying" is not clear (page 5, line 126); what Authors man by check solubility in appropriate solvents? (page 6, line 185);
9./ If there is presence of numbering of carbon atoms in 13C NMR spectra this numerstion should be also shown in some Figure;
10./ Paragraphs devoted to the preparation of stock solutions are not necessary. Preparation of solutions is an obvious step and should be skipped;
11./ Influence of the synthesized compounds on activity of amylase (in experimental is glucokinase) cannot be treated as in vitro antidiabetic activity (this is just rough model);
12./ in a paragraph "2.3.2.2. Protocol of Anti-Diabetic Activity" the differences in absorption (what kind, which wavelength) are taken as % of inhibition?
13./Antiproliferative effects are not anticancer ones;
14./ Results and Duscussion starts from identifying rationale of undertaken studies and thus should be combined with introductory part, and all together consequently shortened;
15./ there is no need for such a detailed discussion of NMR spectra (page 10) since they are rather standard;
16./ I would like to propose to compose supplementary material, where many of the presented data might be moved. This considers figures 6 and 8 (also figures from the section devoted to computer-aided studies), and tables 1, 2, 3, 4, 5, 6 and 7; by the way there is no need to mark compounds 5 at Figures 6 and 8 as U1...U8. The introduction of novel abbreviations results in loss of clarity;
17./ Last sentence in paragraph 3.1.1.1. does not relate to its content at all;
18./ Authors already noticed (line 398, page 12) that the number of studied compounds is to small to draw suitable straucture-activity relationships but then discussed SAR very detaily. All these discussions should be shortened along all the paper;
19./ Figure 10: such high increase of viability (please define the type of cells) is strange and is not an indice of anticancer activity;
20./ Paragraphs devoted to molecular dockings are too spacious. Once more, these are dockings to certain enzymes chosen as models for some physiologic actions but it does not mean that these inhibitors are antidiabetic, antioxidative or anticancer. As seen from docking results the obtained compounds are bound to all the targets in dispersed manner (for example in Fig. 12 C=O moiety is bound in a different manner for each compound). I would like to advice to shorten these paragraphs significantly concentrating on chosen examples. By the way docking results shown in Figures are illegible.
21./ In my opinion paragraph could be skipped without harm to the work;
22./ Paragraph 3.4. has to be significantly shortened or shortened and moved to supporting material.
Reviewer 2 Report
The manuscript entitled Structure-based Designing, Solvent Less Synthesis of 1,2,3,4-tetrahydropyrimidine -5-carboxylate derivatives: A Combined in vitro and in silico Screening Approach describes the synthesis of eight new pyrimidine derivatives and biological and computational studies of their biological activity.
The paper has valuable information but the redaction is very deficient, it is very recommended that a native English speaker revise the redaction, grammar and punctuation marks.
Some examples of many paragraphs to correct:
“This class of heterocyclic synthetic compounds contains many naturally occurring therapeutic drugs like quinine , emetine , procaine , morphine and reserpine shown in figure 1.” It should not be spaces before the commas.
“gemcitabine sulfadiazine HIV inhibitor zidovudine ultrashort acting barbiturates Pentothal antiepileptic drug methylphenobarbital antimalarial drug pyrimethamine antihypertensive agent minoxidil shown in figure 2” Add commas.
“having no byproduct or side reaction an eco-friendly Grindstone technique has been employed.” It should be: …byproducts or side reactions, an eco-friendly…
“The material (chemicals and solvents)” should be The materials…
“no need to further purify.” Should be no need to further purification.
“For the synthesis of required compounds, ecofriendly, one-pot multi-component methodology was used with some modification” should be modifications
Here some observations that I consider necessary to publish the article.
The Abstract is too long and very general, for example there is no need to explain that the characterization has been done using NMR spectroscopy or give a detailed information of the software used. The abstract should summarize the most relevant content of the manuscript and describe in a short description the most relevant results.
In the discussion it should be added a proper scheme of synthesis, adding solvents, temperatures, and yields. Remove figure 5, if the authors consider necessary, it should be part of the introduction, not the discussion.
In the methodology add the frequency of the NMR magnet
a pinch is not a measure unit, describe properly.
In the sentence “re-crystallized with hot alcohol” specify which alcohol.
TLC is not a reliable method to confirm the purity of a compound since it could be mixtures of isomers with very close Rf.
In figure 4, check the subscripts of the formula, yellow letters are illegible.
In “Synthesis of ethyl 4-(fluorophenyl)-6-methyl-2oxo-1,2,3,4-tetrahydropyrimidine-5-carboxylate (5a-f):” it should be …4-(fluorophenyl)-6-methyl-2-oxo-1,2,3,4-tetrahydropyrimidine-5-carboxylate…
In “200mM (0.2M, 6.2055g)” it is not necessary to add quantities inn both mM and M, authors should suppose that the reader can make the transformation. Four digits after the decimal point is unnecessary, leave just one.
In “urea (2a, 0.2M, 3g)/ thiourea and” is missing the quantities of thiourea.
The phrase “200mM (0.2M, 6.2055g) solution of Fluoro-benzaldehyde (3a-f), urea (2a, 0.2M, 3g)/thiourea and ethylacetoacetate (1, 0.2M, 6.507g) were prepared in a china dish and catalytic amount of CuCl2. 2H2O (4) was added to it and grinded together these reagents for 7-10 minutes.” Is not understandable, rewrite adding the word mixture where appropriate.
Many constant coupling values are missing, they should be added in all cases.
In the DPPH assay, specify the quality of methanol used, common methanol is added with antioxidants to prevent degradation, so the results could be compromised.
Before the characterization, discuss the synthesis.
All nmr discussion should be supported by a proper chart to an ease comparison of the values and simplify the text. If data are consistent it is not necessary to describe each value of each compound. When describe values in the text add the unit of the chemical shifts.
Stereogenic is one word.
Round 2
Reviewer 1 Report
Now, after quite significant changes, the paper looks far better. However, there is still a problem with the part devoted to the molecular modeling. This part is overloded with small figures, which do not allow to draw any reasonable structure-activity relations. I do propose to concentrate on most representative examples with shifting most of the material (especially tables describig interactions) to supplementary document (such documents are very useful). I would also reccomend to try to construct something like pharmacohore model for each enzyme representing various physiologic effects - the studied compounds are of significant similarity and that should be posssible. Anyway, some ordering of this part of paper is strongly required.
Author Response
- Now, after quite significant changes, the paper looks far better. However, there is still a problem with the part devoted to the molecular modeling. This part is overloaded with small figures, which do not allow to draw any reasonable structure-activity relations. I do propose to concentrate on most representative examples with shifting most of the material (especially tables describing interactions) to supplementary document (such documents are very useful).
- All the tables were added to supplementary data as recommended, while molecular docking portion was also concise by adding docking view of only lead compounds in all three activities.
- I would also recommend to try to construct something like pharmacophore model for each enzyme representing various physiologic effects - the studied compounds are of significant similarity and that should be possible. Anyway, some ordering of this part of paper is strongly required.
- Thank you for this keen observation and helpful suggestion. I might perform pharmacophore characterization of my synthesized product to get the idea of the final structure-activity relationship, potential of our compounds, but due to time limitation, the complexity of my experiment and costs involved, I can only end up with the Molecular docking, QSAR analysis, MEP and FMO analysis. Next time, we definitely consider your suggestion to do pharmacophore generation as well. It is a very nice suggestion, but we don’t have anyone software for generating pharmacophoric features. We are purchasing software regarding pharmacophoric features generation, will have plans for including this in our next future work.

Round 3
Reviewer 1 Report
Now , after intensive corrections paper is ready to be published.